# Failure Identification Using Model-Implemented Fault Injection with Domain Knowledge-Guided Reinforcement Learning

**DOI:** 10.3390/s23042166

**Published:** 2023-02-14

**Authors:** Mehrdad Moradi, Bert Van Acker, Joachim Denil

**Affiliations:** 1ICT-Department of Applied Engineering Faculty, University of Antwerp, Prinsstraat 13, 2000 Antwerp, Belgium; 2Flanders Make, Gaston Geenslaan 8, 3001 Heverlee, Belgium

**Keywords:** domain knowledge, fault identification, fault injection, reinforcement learning, safety assessment, signal temporal logic

## Abstract

The safety assessment of cyber-physical systems (CPSs) requires tremendous effort, as the complexity of cyber-physical systems is increasing. A well-known approach for the safety assessment of CPSs is fault injection (FI). The goal of fault injection is to find a catastrophic fault that can cause the system to fail by injecting faults into it. These catastrophic faults are less likely to occur, and finding them requires tremendous labor and cost. In this study, we propose a reinforcement learning (RL)-based method to automatically configure faults in the system under test and to find catastrophic faults in the early stage of system development at the model level. The proposed method provides a guideline to utilize high-level domain knowledge about a system model for constructing the reinforcement learning agent and fault injection setup. In this study, we used the system (safety) specification to shape the reward function in the reinforcement learning agent. The reinforcement learning agent dynamically interacted with the model under test to identify catastrophic faults. We compared the proposed method with random-based fault injection in two case studies using MATLAB/Simulink. Our proposed method outperformed random-based fault injection in terms of the severity and number of faults found.

## 1. Introduction

Cyber-physical systems (CPSs) are intelligent systems in which the computational part of the system controls the physical part in order to perform a predefined task [1]. The safety assessment of CPSs is critical, as any failure in the system can cause harm to humans or the environment [2] in some applications such as self-driving cars. To avoid system failure, vendors utilize experimental or simulation-based methods. However, the increasing complexity of CPSs makes safety assessment a time-consuming and costly process for vendors [3].

Based on dependability taxonomy [4], a failure happens if a system’s service under test deviates from delivering the correct service. A service is a sequence of perceivable states at the system’s interface. The deviation from the correct (sequence of) state is called an error. The cause of an error is called a fault.

Safety engineers, among others, are interested in finding faults that cause failures in the system under test. In the rest of this work, we refer to these faults as catastrophic faults. The probability of finding a fault in a system follows a long-tailed distribution, and catastrophic faults usually fall into the tail of this distribution, which means that finding them is complicated and requires a tremendous amount of effort and time [5,6].

One well-known method of safety assessment is fault injection (FI) [7]. With fault injection, safety engineers disturb a part of the system and observe the effects on the total system’s behavior. Therefore, engineers create or simulate a harsh environment for the system [8] to assess the system under test. Fault injection is recommended in functional safety (ISO 26262 standard) [9], and is mandatory in the Standard for Safety for the Evaluation of Autonomous Products (UL 4600) [10] in the automotive industry. In the ISO 26262 standard, fault injection can be used for testing hardware/software levels, diagnostic coverage, and checking the correct implementation of the functional safety requirement and the safety mechanism in different abstraction levels. At the sensor level, test engineers create hardware faults, software faults, and communication faults as test cases for the system under test. They model these faults in detail based on real-world faults in order to accurately assess the safety properties of the system. In the UL 4600 standard, fault injection is introduced as a mandatory step for fault coverage and mitigation. In another standard, entitled Safety of the Intended Functionality (SOTIF or ISO/PAS 21448) [11], test engineers try to find corner-case scenarios by introducing external faults. Identifying potential faults and vulnerabilities in a system improves the safety and reliability of that system.

Traditional fault injection methods are based on exhaustive injection processes [12], random-based fault injection [13], model-implemented fault injection, and abstracting the system under test to eliminate some details [14]. Using previous methods, safety engineers aim to find catastrophic faults that cause a system failure. However, most faults cause masked errors. These masked errors do not further propagate to the external state of the system [4]. This fault-masking decreases the performance and efficiency of fault injection.

(i) Traditional fault injection-based safety assessment methods are not efficient in finding catastrophic faults (in the tail of distribution) and do not scale up to the complexity of future systems. As such, engineers need to produce enormous amounts of data, which is costly and impossible, especially if the system is still under development. There is a need to find an efficient way to deal with data scarcity. Future systems will consist of hardware such as sensors, mechanical and hydraulic components, etc., as well as software for controlling or monitoring hardware via the transferring of data and a high level of interaction. In the automotive industry, vendors are trying to find these hidden failures by testing prototypes on the road with millions of driving miles or by running millions of simulations in simulators and investing a massive amount of time and resources [15,16].

(ii) Traditional fault injection methods are mainly human-driven activities that introduce human errors in fault injection experiments, increasing the time and cost of safety validation. Additionally, many of these systems are composed of complicated components (like heterogeneous systems), and the systems’ implementation details are not adequately understood [17]. The test engineer must choose the injection location, time, and value based on knowledge of the system, such as system architecture, modes, components, communication links, etc. Without this information, the test engineer cannot inject a fault in the proper location, and most of the injected faults would not lead to failure. (iii) The literature has mainly studied simple fault models, for example static faults, meaning that the fault parameters are fixed during simulation, such as stuck-at faults and bit flips [18,19]. In the case of dynamic faults, the fault’s parameters change over the simulation-like noise of the signal. Dynamic faults can cause immense consequences if they are active at a specific point in time with a critical amplitude. (iv) Finally, the safety criteria of a system are often defined over the system under test and its environment. The requirement, “the test vehicle should not collide with pedestrians”, is an example of a requirement that affects both the system under test (the test vehicle) and the actors in its environment (pedestrians). As a result, a safety assessment of the combined system (system and operating environment) is required [20].

This article proposes a method to address the gaps in identifying catastrophic faults. (i) We utilize a machine-learning method to improve the efficiency of model-implemented fault injection. Model-implemented fault injection enables us to frontload the safety assessment to the model level. (ii) The proposed method is semi-automated and requires less user involvement in the safety assessment. (iii) The method applies to dynamic fault types that can be activated and deactivated at any time. (iv) Finally, the proposed method examines the model under test and its environment for fault injection rather than focusing on either one or the other.

To achieve our main goal, we utilize reinforcement learning (RL). Reinforcement learning is a subcategory of machine learning in which an agent learns by interacting with the environment through trial and error [21]. The agent collects data dynamically through this interaction and does not need a static data set. The agent automatically applies actions and controls the fault parameters in the fault injection experiment. In the fault injection experiment, we inject one fault in each experiment based on domain knowledge about important parts of the model. Then, the RL agent quantifies the consequence of the injected fault using a reward function. The reward function determines the robustness of the fault model and whether the injected fault led to a failure, as it dynamically assesses the safety level of the system. If the reward function becomes smaller than a threshold, it indicates that the safety requirement is violated. Besides the reward function, an RL agent has numerous parameters, and its configuration greatly impacts the results of fault identification. Proper setup for fault injection experiments using an RL agent is challenging for safety engineers. As such, the safety engineer needs a proper guideline and workflow to configure the RL agent. We use domain knowledge about the model under test to reach the proper reinforcement learning configuration.

The reinforcement learning agent injects faults into each simulation step and eventually learns to fail the model under test. The agent identifies fault patterns or a sequence of faults that causes a catastrophic failure in the model under test. In essence, the agent learns to fail the system.

This article details model-implemented fault injection for the safety assessment of technical systems. Our main contribution is the reinforcement learning-based method for fault injection that utilizes domain knowledge to set up the different parts of the reinforcement learning agent to find catastrophic faults in the model under test. We summarize the contributions as follows:Firstly, we introduce a reinforcement learning-based method to identify dynamic faults. The reinforcement learning guides the fault injection simulation to detect faults in the system model by injecting faults.Secondly, we focus on extracting high-level domain knowledge. This domain knowledge is based on the model under test (e.g., the model’s architecture, input boundaries, and states), safety requirements, and temporal behavior of the model under test. The domain knowledge helps us to prune the fault space for the RL agent and focus on the most important fault parameters, for example, fault location and fault value boundary. In this way, the RL agent automatically explores the fault space and finds other fault parameters. The model composition impacts fault location and fault boundary value that maximizes the efficiency of fault injection.Thirdly, we provide guidelines to configure different parts of the reinforcement learning agent based on the extracted domain knowledge. We define a technique to shape the reward function using temporal logic and safety requirements.Finally, we validate our approach using two experimental case studies and discuss the limitations and benefits. Our case studies include the Simulink model of adaptive cruise control systems and autonomous emergency braking systems, two important systems in modern vehicles.

In the proposed method, we assume that the domain knowledge given to the proposed method is correct. The proposed method cannot achieve a result with the wrong input data. In addition, to perform our method, the executable model of the system under test is required. Our method can be performed with models at different levels of abstraction and approximation. However, the quality of the results also depends on the quality of the model. Often, complicated subsystems are used in modern CPSs, such as artificial intelligence and machine learning-based subsystems. These subsystems are usually considered black boxes, and the proposed method can help verify them in the interface signals [20]. The proposed method injects faults in the signals within the model under test and can be applied to black-box models as long as signals are observable or externally reachable. Nonetheless, the proposed method can also be applied to an internal fault if the internal signals or internal states of the system are observable. The test engineer has to decide on and adopt the model under test. We also assume that the fault models in the fault library are representative and are modeled correctly. Any small change in the fault model deviates from the result of the proposed method. In addition, we assume that the simulation environment accurately represents the real-world system. Therefore, the simulation result of the faulty model is similar to that of the faulty system in the real world.

In previous work, we implemented a proof-of-concept reinforcement learning agent for model-implemented fault injection and tested it on a single-use case [22]. This paper extends the previous work by (i) using multiple types of reward functions, (ii) applying multiple complex forms of reinforcement learning algorithms, (iii) presenting an extensive experimental evaluation, and (iv) providing a guideline for using reinforcement learning in model-implemented fault injection.

## 2. Background and Related Works

### 2.1. Background

In this section, we describe the fault injection (FI) and reinforcement learning (RL) algorithms, as they are the main concepts of the paper.

#### 2.1.1. Fault Injection

Fault injection is a testing technique in which the test engineer observes the behavior of a system when the system (real or virtual) is stressed in an unusual way [7]. Fault injection is performed on all levels of abstraction, such as in a model, in software (e.g., source code or binary code), or in hardware prototypes. In this paper, we use fault injection at the model level. The advantage of model-implemented fault injection is that engineers can apply the technique in the early phases of system development when the product is just a computer model. Finding errors in the early stages of development reduces the cost tremendously and saves time and effort for the software [23] and hardware prototypes [24,25].

An essential element of fault injection simulation is the fault parameters. Based on [7], faults have three main parameters that create the fault space. The fault dimensions are (i) fault type, which indicates which type of fault should be injected, (ii) fault location, which indicates where the fault should be injected, and (iii) fault activation time, which indicates when the fault should disturb the system under test. Faults are classified into three main categories: (i) transient faults only remain for a short time, (ii) permanent faults remain for a long time in the experiment, and (iii) intermittent faults occur occasionally during the experiment [7,26].

#### 2.1.2. Reinforcement Learning

Reinforcement learning (RL) is a problem-solving framework for problems described as incompletely known Markov decision processes (MDP). The Markov decision process is a general mathematical problem representing an optimal sequence of sequential decisions in an uncertain environment. At each sequence step, the agent takes action and advances to the next state. Negative rewards or positive rewards can be obtained depending on the current state.

The reinforcement learning algorithm, when implemented in an agent, learns by interacting with a dynamic environment during reward-based learning [21]. The environment is the model we want to control or from which we want to learn. The agent aims to find a strategy that generates the maximum reward. This trial-and-error learning process allows the agent to make a series of decisions to perform a task without being explicitly programmed through human intervention. The reinforcement learning agent needs to interact with an environment with the Markov property: given the present, the future must be independent of the past. We can represent this statement mathematically as:(1)$$P\left[S_{t+1}|S_{t}\right]=P\left[S_{t+1}|S_{1},\ldots \ldots ,S_{t}\right]$$

*S*[*t*] represents the agent’s current state and *s*[*t* + 1] represents the next state.

*Reinforcement Learning Structure.* There are five main parts to setting up a reinforcement learning problem: *environment*, *reinforcement learning agent*, *reward function*, *action signal*, and *observation signal*. Figure 1 shows the architecture of RL. The environment is, in our case, the model under test with its surroundings. There are three signals between the *agent* and *environment*: (i) the action signal that controls or manipulates some environmental signals; (ii) the observation signal that reads the state of the system and the most important signals in the environment; and (iii) the reward function that expresses to the agent how well the action value performed.

In addition to the agent, there are two other components within RL. The first is the *policy* that maps observation signals to action signals. The second is the *reinforcement learning algorithm* that updates the policy based on the reward signal, the previous action, and the current observation. This algorithm adjusts the policy at each iteration to collect a higher reward. The reinforcement learning algorithm uses the Bellman equation to discover the best policies and value functions. We know that our policy evolves. Thus, different policies will have different value functions. The optimal value function produces the most value compared to all other value functions.
(2)$$v\left(s\right)=E\left[R_{t+1}+\gamma v\left(S_{t+1}\right)|S_{t}=s\right]$$

The Bellman equation indicates that the value function can be decomposed into an immediate reward (*R*[*t* + 1]) and the discounted value of the successor states (γ).

*Reinforcement Learning Algorithms.* Multiple reinforcement learning algorithms are described in the literature. We can divide reinforcement learning algorithms into two main categories—*model-free* and *model-based*—according to the context used. A model-free algorithm is an algorithm that predicts the optimal policy without using or envisioning the dynamics (transition and reward functions) of the reinforcement learning environment. This contrasts with model-based algorithms, which use the transition function (and the reward function) to estimate the optimal policy. The transition and reward functions are referred to as the environment model.

Furthermore, we can classify the reinforcement learning algorithm into three main categories—*value-based*, *policy-based*, and *actor–critic*—according to the implementation strategy. Value-based agents rely on an indirect policy representation and use only the critic’s network to select their actions. The critic network aims to estimate reward values and modifies the policy based on actions taken, collected observations, and reward values. In each iteration, the critic network tries to reduce errors and accurately predict future reward values. Policy-based agents rely on a direct policy representation that uses only actors to select their actions. Agents that use both an actor and a critic are actor–critic agents. During the training of these agents, the actor learns the best action to take, using feedback from the critic (instead of using the reward directly). In parallel, the critic learns the value function from the rewards to appropriately criticize the actor.

#### 2.1.3. System Requirements and Signal Temporal Logic

System development typically starts with a requirement definition phase. These requirements are the basis for engineers to build and develop a system. The designed system must meet the requirements of the final product. The requirements can be defined by standards, owners, or system users. In this paper, we focus on the safety requirements of the system. We assume that the system is already virtually designed up to a level of abstraction usable for evaluation. As such, we have an executable model of the system.

Often, requirements are defined in natural language, resulting in misinterpretations. However, languages are available to specify the requirements very precisely. Using these languages also makes a safety requirement amenable to automatic analysis. For example, X>Y can be a naive system requirement that compares two signals in the system. Signal *X* must always be greater than signal *Y*. This formula is simply about comparing the values of two signals without any notion of the temporal behaviors of the system. If we want to add temporal behaviors to our formula, a known formalism is signal temporal logic (STL).

Signal temporal logic is a logic language that provides a systematic way to describe temporal and spatial signal properties [27]. It is a language that consists of logical operators (like ¬,∧,∨) together with temporal operators (such as *always* (G), *eventually* (F) and *until* (U)). For example, the formula always[0,30](x≥0) indicates that *during the first 30 time units, the value of signal x is greater than or equal to 0.*

Safety requirements can be formalized in a temporal logic signal form that is quantitative and machine-interpretable. Robustness can be designed over a trace together with a formula (how much a signal satisfies a specification or violates it), and can be used to validate and verify the system requirements [28].

### 2.2. Related Work

For optimizing fault injection, numerous studies are conducted. In these studies, the authors mainly use machine learning techniques, statistics, domain knowledge, historical data, and abstraction techniques to evaluate the system properties to increase the performance or efficiency of the fault injection simulation/experiment. In the rest of this section, we describe each technique in more detail.

#### 2.2.1. Test Automation Methods

Test automation is an important and active research area, especially for industry. Test automation is mainly used in settings that work on complex and large-scale systems, in which the use of traditional testing methods is costly [29]. Hence, they try to (semi) automate testing phases and reach an agile method of system development while keeping the system safe and reliable. Model-based testing is similarly focused [30]. Software engineers try to model the behavior of software based on its architecture and use that model to test the software automatically. For example, they can automatically generate test suits for testing the software [31] using finite-state machines. Each state represents an actual application state in high-level abstraction. These state machines allow us to extract a set of test-case design rules. With these rules, the test engineer can generate a series of application events to satisfy a given set of coverage requirements.

#### 2.2.2. Historical Data

In fault diagnostics, much research has been performed on utilizing historical data [32,33,34]. In [32], the author presented multiple approaches for the design of test plans in order to more quickly measure reliability. For example, by injecting faults into the system based on component failure data, they accelerated component wear-out and measured the reliability of the overall system in normal conditions in an accelerated way. In [33,34], researchers made predictions about faults and failures using information from fault tree analysis (FTA) and failure mode and effect analysis (FMEA). A fault tree provides information on the propagation paths of faults in the system. This information focuses on the most important parts of the system. Additionally, Markov models in [35] and Petri nets in [36,37] were used to reduce testing effort and dependability analysis. The authors constructed a Petri net or Markov model based on historical knowledge about the system, which accelerated further failure analysis.

#### 2.2.3. Machine Learning

Machine learning algorithms are effective in decision-making and are used to handle fault injection experiments. In [38], the authors tried to find failures with a high probability of occurrence using reinforcement learning-based stress tests. They used a stress test for a black-box system in a stochastic environment. The authors in [39] presented a framework that generates and collects more errors in a reasonable time. They used supervised and unsupervised machine learning algorithms to eliminate parameters that did not directly relate to the system’s soft errors. In unsupervised learning, engineers utilize unlabeled data, and they aim to extract patterns from data to group them without any prior knowledge about the structure of the data. However, in supervised learning, the data are labeled, and the algorithm aims to extract features from the data and separate the data into different classes. In [40], the authors proposed a method based on unsupervised machine learning to analyze traces of fault injection automatically. This unsupervised ML detects anomalies and classifies failure mode. In [41], researchers developed a framework called AV-Fuzzer, which uses a search algorithm to inject faults into evolving traffic scenarios to minimize safety. Moreover, researchers used reinforcement learning agents in [22,42,43] to find catastrophic faults using system specifications. The reinforcement learning algorithm can deal with time series data and make sequential decisions about catastrophic faults using dynamic interactions with the system under test.

#### 2.2.4. Using Domain Knowledge

The authors in [44] exploited sensitivity analysis to find the most sensitive traces in a black-box model, which was implemented based on a functional mock-up interface (FMI) [45]. They explored the perturbation’s impact on their use case’s safety requirements and then used that information to inject fault. In addition, the authors in [41] used vehicle dynamics in an abstracting system and performed faster fault injection. In [46], the authors used the fault propagation path for fault diagnostics and identified the root cause of the abnormal situation. Moreover, in [47], the authors analyzed the resilience of a system using an error propagation path.

#### 2.2.5. Statistical Methods

Statistical methods are used for fault space exploration. In [48], the authors developed a domain-specific language called *Scenic* that generates test sets for autonomous cars and robots. These test sets are used to verify, debug, and train machine learning-based systems and increase their reliability. Scenic uses a probabilistic programming language to sample the input distribution. In [49], the authors randomly injected fault and then estimated the confidence level of outcomes. In [50], a framework called *CriticalFault* was proposed. CriticalFault avoids unrelated faults, focuses on high-impact faults, and prunes the fault space dramatically. In [51], some statistical tools were proposed that were utilized for the sensitivity analysis of the system under test and reduced the parameter space for fault injection.

#### 2.2.6. Abstracting of System

Researchers use a simplified system model under test. This allows us to focus on a specific system property, such as real-time behavior, and perform evaluation faster. In [52], the authors aimed to check the fault tolerance mechanism. They used simulations at multiple levels of abstraction, from the hardware level to the software and model levels. To validate a very high-speed integrated circuit hardware description language (VHDL) model, the authors of [53] simulated the register transfer layer to increase fault injection performance. In [54], the authors also used multi-level simulation, focusing on the timing behavior of the system.In addition, in [55], the authors modeled the system as a graph at multiple levels of abstraction to identify system failure via simulation.

All the different methods mentioned above have limitations. In many cases, no data are available from the system under test since the system is still under development. Therefore, supervised learning- and unsupervised learning-based methods are not effective. In addition, in some cases, there is not a sufficient resource to perform numerous simulations. Therefore, methods based on statistics and test automation are not applicable for fault injection. Furthermore, sometimes there is not sufficient information about the system under test. In these cases, the test engineer cannot properly make an abstract model from the system under test or distinguish between important and unimportant parts of the system. Therefore, methods that are based on the simplification of the system and domain knowledge are not relevant.

The proposed methodology combines reinforcement learning, high-level domain knowledge, and test automation. It performs a series of injections inside the system under test and tries to fail the system specification at each step of the simulation. The user can set up fault injection using a high-level fault model, and the reinforcement learning agent learns to inject catastrophic faults via iteration.

## 3. Reinforcement-Based Catastrophic Fault Identification Method

This section introduces a guideline to assist the user in setting up the RL agent for fault identification. The high-level idea of this method is to employ a reinforcement learning agent for model-implemented fault injection to find catastrophic faults in the model under test. Catastrophic faults violate the model’s specifications or safety requirements, and the proposed method semi-automatically identifies them.

The proposed approach requires less user involvement, and it is scalable to a complex model consisting of many components. The underlying neural net in the RL agent can understand the system functionality and control the experimentation. The RL agent injects faults into each step of the simulation and interacts with both the controller and the environment around the controller. Therefore, fault identification is achieved in a broader context that matches real-world case studies. The interaction of the RL agent with the environment does not influence the running scenario of the case study. The RL agent only changes the action signal and monitors the observation signal and the reward value.

The proposed method relies on domain knowledge (specifically model specifications) to set up a fault injection campaign and reinforcement learning agent for fault injection. In the first step, knowledge about the system under test is retrieved to set up different parts of the reinforcement learning agent. Figure 2 shows an overview of the proposed method. Next, the model under test is automatically annotated with the fault model, reward functions, observation signals, and action signals based on domain knowledge. The action signals are connected to the fault model to change the fault parameters during execution.

In the middle of Figure 2, a workflow diagram details the steps of the approach. Each activity in the workflow model shows which artifacts the activity takes as input and produces as its result. As is shown, the first step is manual, the second and third steps are semi-automatic, and the rest of the steps are performed automatically. The workflow’s first step is collecting all the needed information to configure the fault injection experiment. We use different sources of information: (a) fault models, (b) signal boundaries and constraints, (c) model architecture, and (d) safety specifications. This information is used in the subsequent activity to configure the reinforcement learning agent and set up a fault injection campaign. The next step in the workflow is to create the architecture of the fault injection experiment. The agent is connected to the faults that must be injected into the model. Given the observation signals and reward functions, the agent pushes the model into failure by changing fault parameters over time.

After the experiment composition, we have two options: proceeding directly to the training phase or performing hyperparameter tuning. Hyperparameter tuning is an optional step and increases the efficiency of the proposed method by selecting the best internal parameters for the RL agent. The hyperparameter tuning framework searches the parameter space of the RL agent using a state-of-the-art algorithm and seeks an optimal value for each of the internal parameters. This step is timely and can increase the efficiency of the RL agent for the next step. Accordingly, the user of the proposed method decides to either perform hyperparameter tuning or neglect it. Afterward, we move to the training step, where the agent tries to inject faults and fail the system. During training, we log training data and agent parameters. The RL agent considers the entire model under test as a single block box (it only interacts with its environment via some signals), and the internal structure of the model under test, such as internal feedback loops, does not affect the training model. Finally, we validate and visualize the simulation result with the trained agent that fails the system. In the following sections, we describe each part in detail.

### 3.1. Collection of Domain Knowledge

Domain knowledge is used as input for the method. Knowledge must be provided in tables or formulas to be usable for the RL agent. The user collects this information to set up the agent. High-level knowledge helps set up the agent and is also used to prune the fault space and skip irrelevant simulations. To be more precise, we explain each item in more detail as follows:Fault model: A fault is a disturbance in a system that can deviate the system from its correct service. The fault model is a representation of a real-world fault that we use to evaluate its impact. For example, stuck-at-zero and stuck-at-high (the voltage of a power supply) are two common faults in hardware. We can model them in a generic form in such a way that the outcome is similar to a real-world fault. We specify fault models that each have two parameters, such as the amplitude of the fault and the activation time.Signal boundaries: This involves knowledge about the ranges of each signal in input, output, or at an intermediate level. This supports the definition of the input space and avoids simulating invalid conditions and out-of-range signals. It also contributes to having a representative fault. Therefore, the fault value remains within the specified range. Otherwise, the fault value does not mimic real-world disturbances.Model architecture: This defines how the model is built up in terms of high-level functional blocks, how those blocks are connected and transfer data, how the controllers work, how many states the model has, etc. This helps determine those parts where faults should be injected. The selection of a fault injection target is determined by the test engineer. The test engineer can utilize further information about the critical components, system constraints, complexity, and risk to prioritize the target.System specification: This relates to model properties or any metric, such as performance, accuracy, robustness, or dependability (i.e., safety and reliability), that quantifies the model accurately.

We can include more domain knowledge, such as running scenarios and simulation configurations. A running scenario relates to environment/test conditions in the model that is developed. The objective is to better understand what the testing environment is and what the assumption is to define the model’s limits. The simulation configuration provides information for the model simulation, such as solver type, step size, etc.

### 3.2. Reinforcement Learning Configuration

This activity provides configuration tables for the RL agent based on the gathered information. We define the agent’s learning task, including how the agent interacts with the environment and any primary or secondary objectives the agent must accomplish. For this purpose, the engineer has to follow the following steps:Create environment: provide tabular information for making an action signal, observation signal, reward function, and fault model in the model under test to formulate our problem as a reinforcement learning problem. The extended model (model under test with all mentioned signals and models) is considered the environment for the agent.Create agent: define a policy representation and configure the agent learning algorithm. For the agent, one can choose an appropriate agent based on Table 1 and use neural networks (NN) within it. The neural network enables the agent to handle complex environments.

In the following sections, we explain each step in detail.

#### 3.2.1. Define Action and Observation Signal

The RL agent needs signals that connect with the environment in order to interact with it. A (test) engineer is in the best position to gather information and choose the appropriate signals. This information is in the form of a table.

The action signal is connected to the fault model and controls the fault parameters. The RL agent performs different actions through the exploration of the fault space and eventually converges on the desired sequence of actions indicating a catastrophic fault. For each action the agent can make, we specify the following properties:System/subsystem: a subsystem or system of interest in the model for the fault injection simulation.Signal name: the signal name of interest for the fault injection simulation.Fault type: the specified fault model for the simulation.Fault range: a fault range must be specified, allowing the agent to define the boundaries of space exploration. These ranges help us scale the signal to [−1,1], mainly for performance reasons.

The subsystem name and signal name indicate the location of the fault model. The type of fault is linked with the desired fault model that needs to be injected. The fault’s range restricts the RL agent’s action to a valid range. Similarly, we define domain knowledge by observing the model in the way we did for the action signal.

#### 3.2.2. Reward Signals

RL agents make decisions based on a reward value. The goal of the RL agent is to maximize the reward value in each learning step. The reward signal indicates how well the agent performs the task objectives. In other words, the reward assesses the effectiveness of adopting a specific action for a given observation (state). Researchers use a positive reward to encourage specific agent actions and a negative reward (penalty) to discourage others. To create an adequate incentive function, expert knowledge is required. A well-designed reward signal directs the agent to optimize the long-term reward expectation. What makes a well-designed reward is determined by the application and the agent’s goals. Based on best practices, there are some methods to leverage this.

In general, the reward values should be continuous. Continuous reward signals aid in training convergence and produce simpler RL agent network architectures. A discrete reward slows the convergence and may require a more complex network structure. Moreover, a common strategy is to define the reward based on sub-goals. These tiny, positive rewards promote the agent to successfully perform the task [56,57].

If a reward function includes multiple signals, e.g., a vehicle’s position, velocity, and acceleration, we can weigh the signals’ contributions to the reward signal based on their respective impacts [58]. In addition, we can shape reward functions from the control model requirements such as cost functions, safety specifications, performance constraints, and limitations in cases where such specifications are already available. In this way, there is less need for several iterations to shape the reward function. The reward value represents the risk of system failure. The RL agent automatically monitors the reward value to verify the suitability of the action taken during the training phase. In this way, both safety analysis and risk maximizing are performed together. In this paper, we propose three alternatives to shape the reward function. The first alternative is to use the simulation model’s already defined evaluation functions. During testing phases, testing functions might be readily available to observe the model. Engineers usually utilize unit tests and functional tests to perform multiple test cases during development. These tests are usually driven by requirements, and developers validate the system requirements by monitoring signals that express the correctness of system behavior. However, these signals should be appropriately adapted. The signals should be inverted if the routine provides a positive value for a good outcome. Multiplying it by −1 ensures that the agent is rewarded for violating the model (safety) properties of the model. Note that a positive robustness value will result in a penalty for the RL agent. At the same time, a negative number rewards the agent for moving towards negative robustness and violating the equation. Note that these functions should return a distance function of how well or badly the system is acting. If the function returns a Boolean, the agent will have difficulty learning.

The second alternative is to transform the model’s robustness and safety requirements into a logical formula to monitor the observation signals in the model. We use signal temporal logic (STL) as the language to describe the requirements. STL is needed, as the signals are continuous-time signals. However, evaluating a logic formula typically results in a Boolean value (the formula is true or false). Nonetheless, it is possible to calculate a robustness metric for several temporal logics, including STL. If the robustness metric is a positive value, we meet the requirement, and the amplitude of the robustness value indicates how robust the model under test is. However, if the robustness value is a negative number, we violate the model requirement or safety property of the model. Accordingly, the amplitude of the STL indicates how harshly we violated the specification. As the STL also measures the model under the test’s robustness, the robustness information from the STL monitor should also be multiplied by -1 and used as an additional reward function.

The third element alternative is to specify custom reward-shaping based on domain knowledge; therefore, the safety engineer identifies intended signals in the model under test and corresponding weights for each signal (to show the importance) to the RL agent. Using our method, these signals must be represented as follows:Variable/signal: The first element identifies the signal or variable in the model related to the specification.Relation: The second element determines the expected relationship between the signal or the variable and the magnitude. There are two types of relations, *inverse* (I) and *direct* (D). A direct relation indicates that increasing the specified signal must increase the reward value. By decreasing the value of the specified signal, the reward value must be decreased. On the other hand, an inverse relation means that by increasing the value of the specified signal, the reward value must be decreased and by decreasing the value of the specified signal, the reward value must be increased.Strength: To specify the magnitude of the relation (sensitivity), three levels have been defined for I and D: *strong strong* (SS), *strong* (S), and normal indicated by no explicit magnitude relationship. In total, there are six magnitude indicators SSD, SD, D, SSI, SI, and I. In the proposed method, *S* indicates multiplication by ten, and *SS* indicates multiplication by one hundred.Sign: In the last element, the impact of the signal on the reward value is defined. If it increases the reward value, it has a *positive* (P) sign (as a motivation for reward). A *negative* (N) sign decreases the reward value (as a penalty for reward).

In order to quantify the strength, the test engineer can utilize domain knowledge about the system and sensitivity information about the system. Domain knowledge provides information about the importance and impact of the signal on the outcome that can be used here. Besides domain knowledge, the test engineer can run multiple simulations and gain system sensitivity information about each signal. Lastly, to build the custom reward function, we can use the generic formula below that is based on the best practice in [56,57,59]:(3)$$\sum_{row=1}^{row=end}Signal\times Relation\times Strength\times Sign$$

#### 3.2.3. Fault Injection Models

In the proposed method, a domain expert must define the fault models for the existing model under test. The fault models must be parameterized so that the RL agent can decide the value, time, and possible place of the injected fault. These faults are often categorized as *data changing* faults and are added as disturbances to the connections between different parts of the model, e.g., sensors and controllers. The selection of a fault injection target is based on the domain knowledge and priorities defined by the test engineer. If the test engineer specified multiple fault locations, the proposed method injects them one by one starting from a high-priority fault location. In each set of training, one fault in a single location is elaborated. Example fault models are shown in Figure 3. The *switch block* enables the RL agent to switch from the *normal value* to the *faulty value*. This model is a generic fault representation of fault nature, enabling the safety engineer to have a permanent, transient, and intermittent fault. The fault activation section subtracts the fault injection time from the current time (ramp block) to check whether the activation time has elapsed. If not, the output of the adder becomes positive, and the switch block conveys *fault value* to the output.

The two fault models aim to change (the sensor) data and are very similar in activation; however, they differ in how they change data. In Figure 3b, we see that a small noise is added to the signal, while in Figure 3a, the fault value of *port 1* is independent of the normal signal value.

#### 3.2.4. Agent Type Selection

Many RL training agents have been developed, such as policy-based, value-based, actor–critic, model-free/model-based, etc. Model-free techniques are typically superior in reinforcement learning’s current state-of-the-art if an accurate model is not provided as part of the problem formulation. However, model-based agents that train their models for planning face the issue of model inaccuracy, causing instability (the further in the future the agent examines, the more inaccurate the results become). Choosing the RL agent depends on the definition of the case study. The first distinction is based on the action space, whether it is a discrete (e.g., up, down, etc.) or continuous (e.g., traveling at a specific velocity) action. For this purpose, guidelines are offered in Table 1.

**Table 1 sensors-23-02166-t001:** State-of-the-art model-free RL agents.

Agent	Type	Action Space
Q-Learning Agents (Q) [60]	Value-Based	Discrete
Deep Q-Network Agents (DQN) [61]	Value-Based	Discrete
Policy Gradient Agents (PG) [62]	Policy-Based	Discrete or Continuous
Actor–Critic Agents (AC) [63]	Actor–Critic	Discrete or Continuous
Advantage Actor–Critic Agents (A2C) [64]	Actor–Critic	Discrete or Continuous
Proximal Policy Optimization Agents (PPO) [65]	Actor–Critic	Discrete or Continuous
Deep Deterministic Policy Gradient Agents (DDPG) [66]	Actor–Critic	Continuous
Twin-Delayed Deep Deterministic Agents (TD3) [67]	Actor–Critic	Continuous
Soft Actor–Critic Agents (SAC) [68]	Actor–Critic	Continuous

The most popular agents used in the agent structure are deep neural networks due to their ability to encode complicated behaviors, allowing reinforcement learning to be used in situations that are difficult to tackle with standard agents. This aids fault injection in the exploration stage of finding a sequence of faults. We only focus on using neural nets as the structure of the agent.

As in research, we recommend starting with a simple agent. Researchers should utilize more complicated algorithms if they cannot reach the desired outcomes. For example, an appropriate agent to start within a continuous action space is the DDPG. Then we have TD3, an improved and more complex version of DDPG. Next, SAC is a more complex version of DDPG that generates stochastic policies. In addition, PPO has more stable updates but requires more training. We described the DDPG algorithm in Appendix A as an example.

Each RL agent has multiple parameters to configure the agent and neural network in the agent [69], as well as training parameters. These hyperparameters have an enormous impact on the training result. In research, it is recommended to start with default hyperparameters or to find an appropriate RL agent in a similar application and utilize its hyperparameter. However, this method is not likely to find a pre-trained agent. Hyperparameter optimization is a solution to tackle this problem. Hyperparameter optimization techniques are mainly based on random and grid searches to explore the parameter space, and then they provide a set of values that increase the RL agent’s efficiency.

### 3.3. Experiment Composition

During the composition of the model, the different components of the experiment are combined. This includes the agent–environment interface and the environment model that consists of the model under test in the running scenario. For reasons of run-time performance and generality of the approach, we convert the agents’ environment model (which is the Simulink model of our use case with a fault model, reward signal, action, and observation signal) to a functional mock-up unit (FMU). An FMU is a black-box unit in the functional mock-up interface (FMI) standard [45]. The FMI is a standardized interface for developing complicated cyber-physical systems in computer simulations. The FMI standard allows for the exchange of models between different platforms and enables us to be independent of Simulink software and any other tool. This makes the proposed approach generic, and the proposed method can be applied to a wider range of tools and applications.

The architecture of the model, together with the reinforcement learning architecture and configuration, need to be combined into a single executable model. In our experiment in Section 4, we perform model composition manually; however, it can be automated via model transformation techniques. More information on composting models using model transformation techniques can be found in [70].

Finally, we connect the exported FMU model to the RL agent via a wrapper. This wrapper allows the RL agent to interface with the FMU in a standardized structure. If multiple fault locations or fault types are specified, the RL agent performs space exploration serially. It starts with the first location and injects the predefined fault type, considering its boundary. Then, it moves to the second one and performs all simulations one by one.

### 3.4. Hyperparameter Tuning, Training, Data Processing, and Visualization

Once the reinforcement learning setup is ready, the training phase commences. Algorithm 1 details implementation of the proposed method. First, the user chooses to perform hyperparameter tuning or to use the default parameters of the RL agent. In the case of performing hyperparameter tuning, the proposed method loads all possible parameters and then returns the best parameters. Next, it starts training using gained parameters in the previous step. In training, the proposed method advances step by step in both the RL agent and the environment FMU until a predefined *timestep*. During training, the RL agent explores the fault space to find an optimal sequence of faults that fail the model under test. Training data are recorded for each trained agent so that performance can be evaluated. The trained agent is then selected for deployment through the fault-injection method [71]. Finally, we can see the results in tabular format and in rendered graphs.
**Algorithm 1** Algorithm of the proposed approach.     *Rewards* = [“Specification”, “Custom”, “STL”]2:  *RLAlgorithms* = [PPO, SAC, A2C, DDPG, TD3]     Set the *TimeSteps*4:  **for**
*Reward* in *Rewards* **do**           **for** *RLAlgorithm* in *RLAlgorithms* **do**6:              Env ← FMU of the RL environment with *Reward*                  **if** HPT==True **then**8:                   *param*←HPT(*RLAlgorithm, Reward, Env*)                 **else**10:                 *param*← default *RLAlgorithm* parameters                 **end if**12:            Reset *Env*                 TRAIN(*RLAlgorithm, Env, param*)14:            Save the RL agent and execution time                 Inject the critical fault16:            Save the outcome                 Render the RL agent18:      **end for**       **end for**20:  **function**
TRAIN(*RLAlgorithm, Env, param*)             RLAgent ← *RLAlgorithm* with the *param*22:        **while** end of *TimeSteps* **do**                  Advance one step in RLAgent24:             Normalize action signals                  Advance one step in *Env*                                                          ▹ DoStep in its FMU26:             Normalize observation signals                  Assign the observation sig. and reward to RLAgent28:             Save reward, observation and action signals            **end while**30:  **end function**       **function**
HPT(*RLAlgorithm, Reward, Env*)32:        params ← parameters of *RLAlgorithm*             **for** param in params **do**34:              Reset *Env*                   Train *RLAlgorithm* with *param, Reward, Env*36:              Save the reward value             **end for**38:        **return** *param* with the maximum reward value       **end function**

## 4. Experimental Study

This section briefly explains our case studies. We use our case studies to validate our approach. Then, we elaborate on our setup to validate and test the proposed method. Finally, we illustrate the result of the proposed method.

### 4.1. Case Studies

We used two case studies to evaluate our method: an adaptive cruise control (ACC) system and an autonomous emergency brake (AEB) system. ACC will be used as a motivating example for domain knowledge extraction and application of the method in Section 4.2.

#### 4.1.1. Adaptive Cruise Control

ACC is a driving assistant system that controls vehicle speed in response to conditions on the road [72]. We used MATLAB/Simulink ACC in [73] as an initial case study. This case study contains an ACC model of an *ego vehicle* that follows a *lead vehicle* as shown in Figure 4.

In this control algorithm, there are two vehicles, named the *leading vehicle* and the *ego vehicle*, as shown in Figure 4. ACC is implemented in the ego vehicle that follows the lead vehicle. The lead vehicle follows a predefined speed profile in different scenarios. ACC has two basic modes: *speed-control* mode and *spacing-control* mode. In speed-control mode, the ACC tries to reach the predefined speed specified by the driver. Therefore, the controller increases the speed while maintaining a safe distance. When the relative distance between the lead vehicle and the ego vehicle is smaller than the safe distance, the ACC switches to spacing-control mode. The controller decreases the velocity of the ego vehicle to increase the relative distance and avoid a crash. The ACC calculates the relative distance from and velocity of the lead vehicle using sensor data from the ego vehicle, as shown in Figure 5. Then, the ACC calculates an acceleration signal that changes the speed of the vehicle.

The ACC system is safety critical, as any failure in this system can lead to an accident and could endanger human life. The ego vehicle must maintain a *safe distance* to avoid accidents by slowing or stopping when the leading vehicle performs a slowing or stopping maneuver. An example safety specification is given as follows:(4)SafetySpec.:safe_distance≥relative_distance;

In Equation (Equation 4), the minimum safety spacing for the longitudinal vehicle is the essence of the safety requirement. This article uses a simplified safe distance that is a constant value during the simulation. If the relative distance is equal to or greater than the safe distance, we are in a safe driving situation.

#### 4.1.2. Autonomous Emergency Braking

Autonomous emergency braking (AEB) is an active safety system that helps drivers avoid or minimize collisions with vulnerable road users or other vehicles. Autonomous emergency braking systems increase safety by (i) avoiding accidents by recognizing dangerous situations early and alerting the driver and (ii) reducing the collision speed to decrease the severity of unavoidable collisions. In this article, we use the AEB model with sensor fusion taken from MathWorks in [74].

The AEB model contains a vision sensor and radar sensors to identify the potential of a collision with the lead vehicle (the vehicle in front of the ego vehicle). This model represents the functionality of a real-world AEB.

The controller has a forward collision warning (FCW) subsystem that warns the driver of an imminent collision with a leading vehicle. In addition, the controller calculates the time to collision (TTC) with the lead vehicle. If the time for TTC is less than the calculated FCW, the FCW warning is activated. In this situation, if the diver does not break in time, the AEB model prevents or mitigates the accident independently of the driver. The AEB model applies cascaded braking, as shown in Figure 6. This figure shows that multi-stage partial braking followed by full braking occurs to reduce the vehicle’s speed.

Autonomous emergency braking is also a safety-critical system and must be reliable. The safety specification of this system is to not collide with the lead vehicle. In case of an accident, the AEB must decrease the vehicle’s velocity. Therefore, the severity of the accident is minimized. Further information about the AEB model can be found in [75].

### 4.2. Domain Knowledge for ACC

This section guides the user on how to extract domain knowledge in detail. This knowledge will be used to set up the RL experiment in Section 4.4. In this section, we use ACC as a motivating example to show how to apply the proposed technique in a case study.

#### 4.2.1. Define Action and Observation Signal

For the ACC case study, the following domain knowledge is collected:Signal boundaries: the relative distance (relative distance = position of ego car − position of lead car) is between 0 and 200 m; the velocity is between 0 and 30 m per second; and the acceleration is between −3 and 2 m per square second;Model architecture: two models define the control algorithm: a *spacing control* model and a *speed control* model; there are two sensors such as *acceleration* and *radar*; and there are two important signals such as *velocity* and *relative distance*;System specification: Safetydistance≥Relativedistance androbustness=Relative_distance−Safe_distance.Running scenario: at the beginning of the simulation, the acceleration of the vehicle is increased and then decreased;Simulation configuration: a fixed-step solver is used with a step-size solver of 0.1 s.

According to the above information, we form Table 2. This table shows our case study’s domain knowledge for action signals. Action signals control fault parameters, which are the link between the RL agent and the fault model (as shown in Figure 7). The first two columns of Table 2, indicate the fault location. These signals are part of the model under test and are those signals to which fault injection actions must be applied. In this paper, for simplicity, we only simulate fault in a single location in the model under test in the *velocity* signal (first row of the table). This signal is the output of the *ego vehicle* stated in the first column. Therefore, the RL agent tries to change the velocity signal data in all time steps. The fault type in Table 2 refers to a fault model that must be defined and added to the model under test.

Table 3 shows our case studies’ domain knowledge for choosing observation signals according to the information in the model architecture. Based on this table, we have chosen all four signals (shown in Figure 7) as observation signals.

#### 4.2.2. Reward Shaping

We use domain knowledge to shape three different reward functions in this section.

*Specification-Based Reward.* According to the safety specifications, we can define a simple reward function as follows: Reward_Value=Safetydistance−Relativedistance

*STL-Based Reward Function for the ACC.* The robustness of the case study is modeled using STL as follows:(5)Robustness=G[0,30](Relative_dist.−Safe_dist.)

The robustness formula defines the difference between relative and safe distance during simulation monitoring (between 0 and 30 s). If this difference becomes greater, the ACC becomes more robust. To convert robustness into a reward function, we use the Reward_Value=−Robustness formula. In this way, the RL agent is motivated to lower the robustness and fail the ACC.

*Custom Reward Knowledge for the ACC*. Table 4 represents the domain knowledge for reward shaping of the ACC case study. A safety specification is defined as shown in Equation (Equation 4), which depends on the relative distance. Hence, the user defines the *relative distance* as a variable/signal. Another variable is velocity, as the engineer aims to motivate the vehicle to move faster. In addition, engineers specify *time* as a penalty in the table so that the RL agent is motivated to fail the model faster. As it is known that the relative distance has a profound impact on the reward value, the (test) engineers specify *S* in the second column. This S indicator will be translated into multiplication of the signal by a factor of 10. We reached this number based on practical experience. In addition, the safety engineer defines *D* here to show the direct relation. It indicates that a higher reward value accrues if the relative distance increases. In the last column, the *P* sign indicates that the reward value will increase for the given relation, and the *N* sign does the opposite. Therefore, the custom reward is as follows: CustomReward=−10×(Rel.Dist.)+Velocity−Time.

#### 4.2.3. Fault Injection Models

We use the fault model and velocity signal shown in Figure 3a. In this model, the agent controls the fault’s activation time and amplitude in each simulation step.

#### 4.2.4. Agent Type Selection

As the reward signal is continuous, the agent’s configuration is performed accordingly. In the section on the experimental study, we use different configurations.

#### 4.2.5. Experiment Composition

Figure 7 illustrates the composed experiment model. The original model is marked in brown. Faults injected into the model are marked in red, whereas green parts are added to connect the model under test to the RL agent at the bottom. The proposed method automatically uses the given information (based on domain knowledge) to build the experimentation model.

#### 4.2.6. Training, Data Processing, and Visualization

In the training phase, the agent injects faults into the velocity signal, and during iteration, it tries to fail the ACC model. Training data and trained agents are recorded and analyzed for the list of successful agents. Finally, we connect successful agents to the ACC and validate and visualize the ACC behavior. All results are represented in Section 4.4.

The next section will demonstrate the experimental setup to execute fault space exploration.

### 4.3. Experimental Questions and Experimental Setup

In this experimental study, we try to answer two questions: (1) How does the configuration of the experiment (such as agent type and reward type) impact the results of finding faults? and (2) How does the reinforcement learning-based method compare to a random-based fault injection method?

We answer the first research question by setting up several experiments. For each use case, we vary (a) three reward functions based on safety specifications, signal temporal logic, and a custom reward; (b) five different reinforcement learning agents (SAC, DDPG, A2C, PPO, and TD3)—since they are state-of-the-art agents for continuous decision making, they are categorized as actor–critic and deep RL; and we (c) allow or disable hyperparameter tuning. As such, we have 3×5×2=30 experiments. Additionally, we used different fault models in our case studies to test their impact on our proposed method. We utilized the change in data as a fault model for ACC and the noise as a fault model for AEB.

We used random-based fault injection (RBFI) as the baseline to answer the second research question. Random-based fault injection uses a uniform distribution and selects a value for each simulation step. In random-based fault injection, fault parameters are randomly selected within a defined range in the action table. To compare our method with random-based fault injection, we use the following metrics in our study: (a) the total number of catastrophic faults, (b) the severity of found faults, and (c) execution time. The severity of the found fault is based on safety violation amplitude. For both the ACC and AEB use cases, we consider −1×robustness as the severity of the faults.

Hardware configuration has an enormous impact on the method execution time. The results of the experiments are obtained using a personal computer with 24 cores (AMD Ryzen 3970x CPU), 32GB RAM, and a 4GB GPU. For the reinforcement learning agent, we use Python v3.8, OpenAI Gym v0.21.0 [76], Optuna v2.10.0 [77], and Stable-baseline3 v1.3.0 [78]. All experimental results, including the models of both case studies, Python code, and recorded data, are publicly available in [75]. The default hyperparameters for each algorithm are shown in Appendix B.

### 4.4. Results

In this section, we demonstrate the result of our method in both the ACC and AEB case studies. Table 5 shows the training time for the proposed method in both case studies. In each reinforcement learning experiment, we iterated 100 times (episodes). The proposed method performs time-consuming training. It needs more time in the case of hyperparameter tuning (HPT), since it needs to try numerous parameters in the RL agent. One iteration in random-based fault injection (RBFI) is fast, but we tremendously increased the number of simulations to ensure that the RBFI method takes the same execution time. Accordingly, we had two modes for RBFI. The first mode was *small* mode. We repeated the simulation almost 70,000 times (almost six hours of execution time) to make its results comparable to reinforcement learning experiments without hyperparameter tuning. The second mode was the *large* mode. We increased the number of iterations to almost 450,000 times to reach almost equal hours to reinforcement learning with hyperparameter tuning.

Table 6 illustrates that the proposed method in the ACC case study found ten critical faults both in cases using hyperparameter tuning and without HPT. However, the RBFI in small mode was unable to find any catastrophic faults, and in large mode it found one fault. In the AEB case study, the RBFI in small mode could not find any fault but found three faults in large mode. The proposed method in the AEB discovered one fault when it did not use HPT, and it found six faults in the case using HPT. Furthermore, in the ACC case study, hyperparameter tuning did not increase the total number of catastrophic faults, while in the AEB, it was effective and could increase the total number of found faults.

In both case studies, the execution time of reinforcement learning without HPT and RBFI in small mode was equal; however, reinforcement learning outperformed RBFI. In addition, the execution time of reinforcement learning with HPT was equal to RBFI in large mode, and Table 6 shows that the proposed method again outperformed RBFI.

An example of the found fault is shown in Figure 8, which shows a catastrophic dynamic fault that failed the model specification. We can see that the fault value and activation reached a maximum value at two points during the simulation. This fault pattern increased the reward value, meaning that the RL agent converged on its goal. In Section 4.5, we will observe the result of injecting such a catastrophic dynamic fault on the vehicle in the ACC.

Table 7 represents the maximum severity of the faults in RBFI and the proposed method. The severity value indicates the severity of missing specifications in both ACC and AEB between the two cars. In the ACC case study, the severity level of found faults using the proposed method was more prominent than the severity of found faults in RBFI; however, in the AEB case study, RBFI found faults with higher severity than the proposed method, but the difference is negligible.

Table 8 shows the total number of faults found per each RL agent’s algorithm in two cases using the default hyperparameter (HP) and tuned hyperparameter (HPT). In the ACC case study, when we utilized the default hyperparameter, A2C was more effective than others, while if we used a tuned hyperparameter, the TD3 algorithm worked better. TD3 is an advanced form of DDPG, and it can handle a more complex environment. Except for DDPG with the default hyperparameter, we can see that all algorithms could find catastrophic faults in ACC cases, either with the tuned hyperparameter or default hyperparameter. In the AEB case study, A2C is also the best performing algorithm, and hyperparameter tuning increased its effectiveness. We can see the effectiveness of hyperparameter tuning on TD3 and DDPG; without hyperparameter tuning, they could not find any fault.

More information about the effect of hyperparameters on reward types in ACC, and the effect of reward types on the RL algorithms in ACC, can be found in Appendix C.

### 4.5. Validation

We attached the trained agent to the environment on the ACC use case to validate its result. Figure 8 represents the action signal, such as the fault’s value and activation time, and the reward value to a trained agent that found a catastrophic fault. Discovered catastrophic fault creates a failure for the ego vehicle in ACC. Vehicle behavior is presented in Figure 9. We can see that the vehicle’s acceleration is above zero, so its velocity eventually increases until 30 s, when safe distance equals relative distance. Then, the acceleration value becomes −2.5, meaning that the vehicle wants to slow down, while the safety specification becomes a negative number. Meanwhile, the reward value increases and becomes a positive number at the end of the simulation time.

### 4.6. Threads to Validity

Following the criteria given in [79], we discuss the challenges to validity in this section. *Construct Validity.* In this experiment, we simplified the severity level of fault by considering the magnitude of violating the safety requirement; on the contrary, in the real world, severity has a broader meaning. In addition, we limited the experiment to one model requirement, which is a safety requirement. By extending the simulations, some injected faults may not influence the defined safety requirement, but they can influence the accuracy of the intended function. For simplicity, we ignored those faults and focused on one model requirement.

*Internal Validity.* The experimental result depends on the underlying hardware or software, such as a toolkit, CPU, GPU, predefined domain knowledge, and a number of iterations of the experiments. All experiments are run on the same configuration, and each is run three times. In addition, we used both the safety specifications and extra information about the model under test for shaping the custom rewards. Different developers may select different types of information based on their perceptions. This selection affects the result, as the reward function is vital to exploring the RL agent. It can impact the accuracy and efficiency of the agent. Furthermore, the fault model is vital, and selecting a different model generates different results. Therefore, fault models and their parameters must be selected on the basis of real-world phenomena.

*External Validity.* In order to check the generalizability of the proposed method, we applied the method to two case studies. The proposed method has adequate results compared to random-based fault injection. In each case study, we used different fault models with different settings and observed that the proposed method achieved valid results. Note that the proposed method needs to be configured differently for different case studies, and safety engineers cannot apply the trained agent from one case study to a different case study. To mitigate this threat, performing hyperparameter tuning is essential to the workflow. Additionally, the complexity of both case studies is almost equal, and our results might not hold for highly complex case studies within a few hours of simulation. However, the RL agent, with extra time and iteration, can reduce this thread.

## 5. Discussion

Our experimental study shows that (a) the configuration of the experiment impacts the results of the proposed method and (b) the proposed method can find catastrophic fault in a semi-automatic setup and outperform random-based fault injection. The proposed method combines fault injection, reinforcement learning, and high-level domain knowledge. The RL agent utilizes domain knowledge to quickly and effectively find catastrophic faults in the system model. Domain knowledge helps in configuring the reinforcement learning and fault injection campaign. The proposed method discovered catastrophic faults in multiple case studies and outperformed the random-based method. However, the proposed method is time-consuming compared to random-based fault injection in a single-point fault injection simulation.

In the proposed method, each reinforcement learning agent focuses on finding high–severe catastrophic faults rather than the number of catastrophic faults. It aims to increase the severity of the fault in each iteration. The user needs to repeat the training, adjust the method’s setting, or add known faults as a penalty to reward function for finding more faults. The proposed method has the following benefits: *No need for static data.* The agent can learn with dynamic interaction with the environment and does not need any data set. This enables the proposed method to work in a system model with no experimental data available, while machine learning-based methods need a considerable static data set for training.

*Suitability for the complex environment.* The agent has a high controller ability and can handle many parameters in uncertain and nonlinear environments. It allows the proposed method to explore a broad space and converge on an outcome. In contrast, statistical methods cannot handle such complicated problems, as a system operates in a complex stochastic environment, and a catastrophic fault follows a concrete pattern [38].

*Generalizability and flexibility.* The agent needs a reward function, action signals, and observation signals to interact with the environment. Therefore, if the case study is observable (accessible) for the agent, one can use the proposed method. This method can be applied to multiple levels of abstraction, from the system’s model to software and hardware. It can also be applied to the black-box system if the essential signals are provided. On the contrary, flexibility is low in methods based on historical data, domain knowledge, machine learning, and system abstraction. They cannot be extended to black-box systems or can only be applied to the system under test at a specific abstraction level.

*Extendability.* The proposed method can be extended to other applications, such as scenario-based testing, fault diagnosis, robustness analysis, and dependability analysis. Finding unknown faults is a fundamental problem in many mission-critical and dependable applications. Nonetheless, except for statistical methods, all the described methods in Section 2 are designed for a narrower scope, and extending them to a broader scope requires more effort.

*No need for detailed domain knowledge.* The proposed method benefits high-level domain knowledge to configure the agent. The detailed nature of the system is not necessary, and having insight into the functionalities of a system under test is sufficient. All of the state-of-the-art methods are discussed in Section 2, except that statistical methods need detailed knowledge about the system under test and labor that increases the cost and makes fault identification error-prone.

The proposed method has the below limitations despite its benefits and effective results:High computation cost in training and tuning: The agent requires simulation trials to reach the appropriate hyperparameter and iteration in training to converge on the desired result.Appropriateness of the domain knowledge: The test engineer needs accurate and sufficient information to set up the agent and framework. The agent will face an enormous or wrong fault space if the data are incorrect or abstract.Dependency of result to toolkits, hardware, and the number of iterations: The proposed framework’s result varies if one uses a different CPU, GPU, or software, especially in terms of execution time.No performance guarantee, and training may not converge: Discovering dynamic faults depends on the proper agent and experiment configuration. Therefore, if the agent could not find any fault, it does not mean there is no possible fault. Some adjustment and iteration of the agent leads to outcomes.

Some of the above limitations originate from the nature of machine learning. For example, most machine learning algorithms are computationally intensive, and there is no guarantee of reaching the desired result. Hyperparameter tuning can help achieve desired outcomes but adds a computational cost. The experimental results in Section 4 illustrate that hyperparameter tuning increases the total number of faults and execution time. Moreover, hyperparameter tuning decreases the framework’s dependency on different use cases. By switching to a different case study, hyperparameters need to be adopted that overcome this obstacle.

In the scope of testing and system debugging, there is always a trade-off between the level of domain knowledge and the performance of methods. Detailed domain knowledge improves efficiency and execution time, but it requires the involvement of experts and a lot of labor before running the experiment. The proposed method is more focused toward involving less labor; therefore, the agent needs to explore a broader space, leading to a long execution time.

Another point is that debugging the system is subjective. There is no answer to how many faults remain. Testing all data values is impractical, and 100% coverage is not fully tested in the best case. Moreover, positively passing tests are not perfectly safe and bug-free. This method can discover faults one after another, and repeating them makes the final system more reliable.

The proposed method is based on deterministic safety requirements; however, the safety requirements can be probabilistic. For example, the likelihood of a certain type of crash occurring must be below a certain percentage. In this case, the test engineer can still define the safety requirement in a deterministic manner to encourage the RL agent to find catastrophic faults that violate the safety requirement. In this manner, the catastrophic faults found fail both the deterministic and probabilistic safety requirements.

System hierarchy and composition of the system can impact the fault injection method. When designing a system (e.g. following the V-model or other model-based system engineering workflows), in the early stages, engineers can use the models to discover different catastrophic faults in advance. This information flow can be used as domain knowledge throughout the workflow (and further detailing) of the design to better conduct fault injection experiments. When transferring to a more silo’ed design phase of the subsystems and components, the system under test changes to the subsystem or component. Similarly, information about this specific domain can be employed as domain knowledge in subsequent experiments. When integrating, the opposite should be done. Information about faults can propagate upwards to the system level and be utilized to identify catastrophic faults at that level. This introduces some interesting challenges as to how to automatically propagate this information between the different levels of abstraction.

Furthermore, multiple-point fault injection can be studied using the proposed method. In this paper, single-point fault injection has been studied in case studies, but in the real world, multiple-point faults can accrue. In this case, the agent can control various faults at different locations in the system under test using multiple action signals. Therefore, the agent needs to explore the larger fault space to find a combination set of dynamic faults.

Extra information such as sensitivity information on signals, historical test data, results of fault tree analysis (FTA), and failure mode effect and critically analysis (FMECA) is extremely beneficiary as is used for pruning the fault space and achieving a more accurate configuration of the reinforcement learning agent in a later stage.

## 6. Conclusions

We proposed a semi-automatic method to find dynamic catastrophic faults in a model under study. In this method, we used a reinforcement learning agent to control a fault injection experiment and high-level domain knowledge to configure different parts of the reinforcement learning agent and the fault injection campaign. We provided a guideline for users to use the method and correctly set up fault injection experiments. We compared the result of the proposed method to random-based fault injection in two case studies, and the proposed method outperformed random-based fault injection in terms of severity and the number of found faults. In addition, our experiments showed that the result of the proposed method depends on the selected reinforcement learning agent and the reward function.

## Figures and Tables

**Figure 1 sensors-23-02166-f001:**
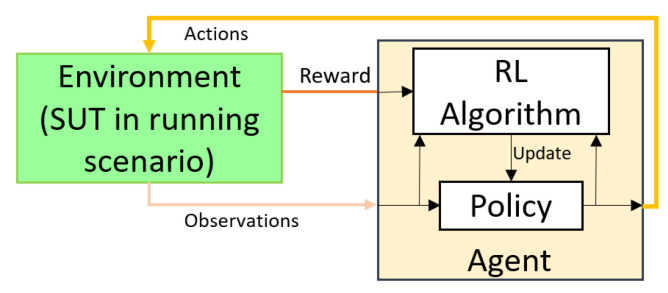
The reinforcement learning algorithm’s block diagram.

**Figure 2 sensors-23-02166-f002:**
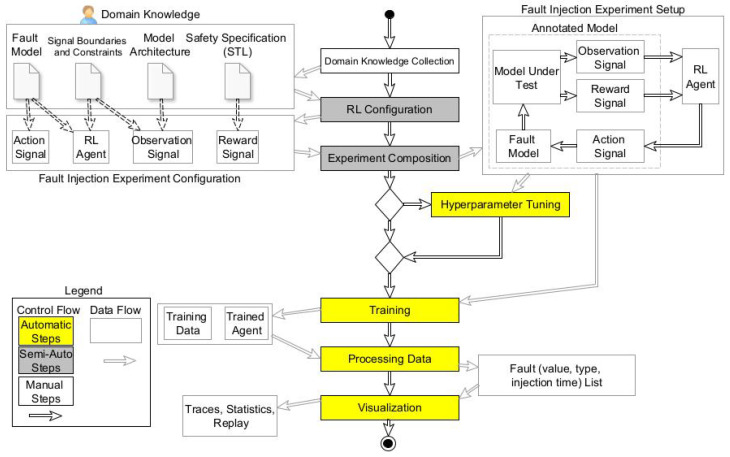
Overview of the proposed method.

**Figure 3 sensors-23-02166-f003:**
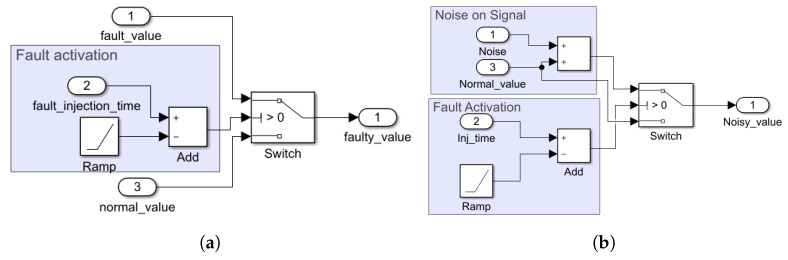
Fault models. (**a**) Fault model of data changing; (**b**) fault model of noise addition.

**Figure 4 sensors-23-02166-f004:**
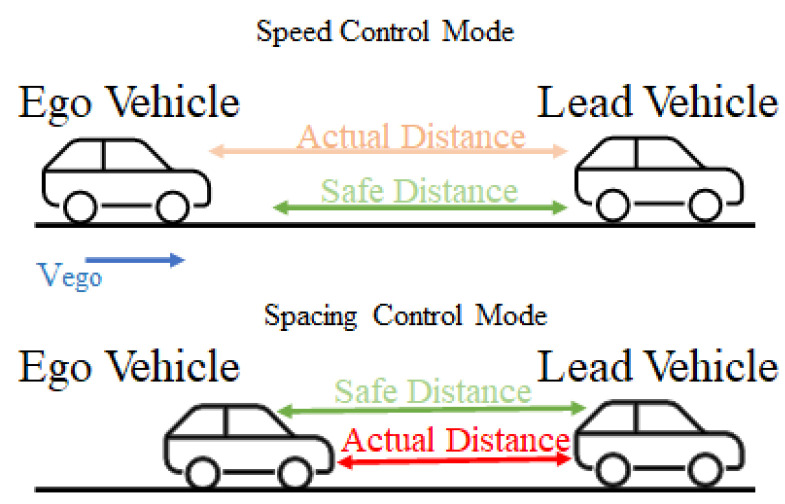
Adaptive cruise control-model operational modes.

**Figure 5 sensors-23-02166-f005:**
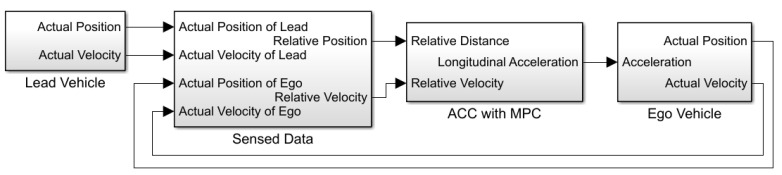
Adaptive cruise control model.

**Figure 6 sensors-23-02166-f006:**
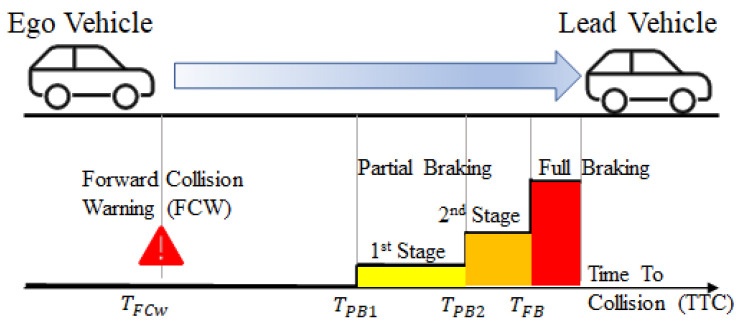
Autonomous emergency braking model modes.

**Figure 7 sensors-23-02166-f007:**
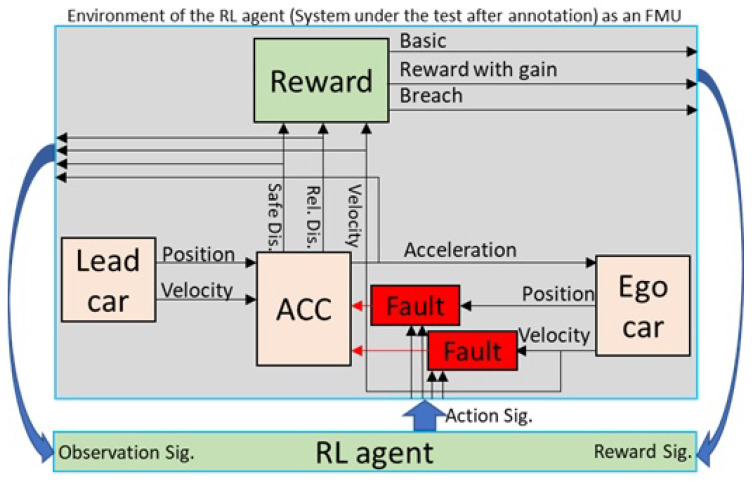
Fault injection setup for ACC using RL.

**Figure 8 sensors-23-02166-f008:**
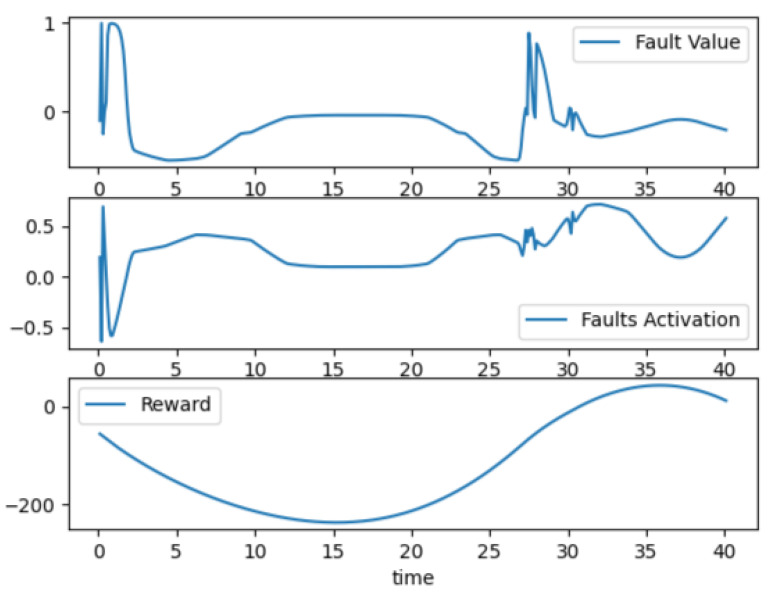
Fault value and activation time of trained agent.

**Figure 9 sensors-23-02166-f009:**
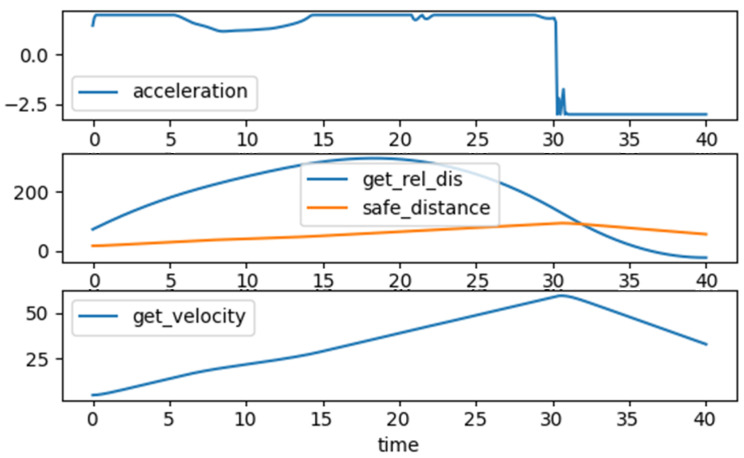
Result of the FI with the trained agent on ACC.

**Table 2 sensors-23-02166-t002:** Domain knowledge for action signal.

System/Subsystem	Signal Name	Fault Type	Fault Range
Ego Vehicle	Velocity	Data change	[−3, 2]
Sensor	Radar	Noise	[−1, 1]
Sensor	Acceleration	Noise	[−2, 2]
Controller	Relative Distance	Data change	[0, 200]

**Table 3 sensors-23-02166-t003:** Domain knowledge for observation signals.

System/Subsystem	Signal Name	Range
Controller	Safe Distance	[0, 50]
Controller	Relative Velocity	[0, 200]
Sensor	Longitudinal Velocity	[0, 60]
Sensor	Lateral Velocity	[0, 10]

**Table 4 sensors-23-02166-t004:** Domain knowledge for reward shaping.

Variable/Signal	Relation and Strength	Sign
Relative Distance	SD	N
Time	D	N
Velocity	D	P

**Table 5 sensors-23-02166-t005:** Execution time of the experiment.

Use Case	RL without HPT	RL with HPT
ACC	5.8 h	114.8 h
AEB	5.9 h	155.7 h

**Table 6 sensors-23-02166-t006:** Number of found fault.

Use Case	RL without HPT	RL with HPT	Small RBFI	Large RBFI
ACC	10	10	0	1
AEB	1	6	0	3

**Table 7 sensors-23-02166-t007:** Maximum severity of the found fault.

Use Case	RL without HPT	RL with HPT	Small RBFI	Large RBFI
ACC	118.5	136.1	-	8.28
AEB	1.34	1.34	-	1.92

**Table 8 sensors-23-02166-t008:** Comparison of RL agents in finding faults in ACC.

Use Case	SAC- Default HP	SAC- with HPT	DDPG- Default HP	DDPG- with HPT	PPO- Default HP	PPO- with HPT	TD3- Default HP	TD3- with HPT	A2C- Default HP	A2C- with HPT
ACC	2	3	2	0	1	2	1	4	4	1
AEB	0	0	0	1	0	0	0	1	1	4

## Data Availability

All simulation materials are publicly available in [75]. The simulation model of case studies, the software code, and the results exist there.

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
