# Peer review of "Failure Identification Using Model-Implemented Fault Injection with Domain Knowledge-Guided Reinforcement Learning"

_sensors, 2023, doi:10.3390/s23042166_

Round 1

Reviewer 1 Report

The paper entitled “Failure Identification using Model-Implemented Fault Injection with Domain Knowledge-Guided Reinforcement Learning” has focused on the safety evaluation of cyber-physical systems and proposed a new approach for fault injection based on reinforcement learning (LR). In addition, the proposed method has been compared with a traditional random-based fault injection and the results were promising. 

Thanks to the author for writing a high quality journal paper. The following comments can be addressed to improve the paper and make it stronger.

1-      In section 2.3.1, it would be good to discuss briefly the state-based models like Markov, Semi-Markov and Petri Nets as well. In section 2.3.2, you have mentioned “Markov models in [35] and Petri nets are [36,37] are used to reduce the testing effort and dependability analysis.”  However, I think they should be considered in section 2.3.1 not 2.3.2. Because the way the Markov and PN is used in Safety evaluation is actually based on historical data. Models like Hidden Markov Model (HMM) can be considered for Machine Learning as they has training procedure.

2-      In section 2.3.2, also, the safety of ML algorithms themselves should be briefly discussed.

3-      Figure 2 was very helpful to understand the idea behind the proposed method. However, it was a bit vague how the hyper parameter tuning is going to be performed. Could you please elaborate on that?

4-      In section 4.4, it is recommended to compare the results with other types of RL as well. For example, actor-critic and Deep RL could also have better results.

Generally, the paper is well-written and has novelty. It is recommended to be accepted after a revision.

Author Response

Dear Reviewer,

We take this opportunity to express our gratitude for your constructive feedback and helpful comments on improving our manuscript.
We have carefully studied the comments and corrected them in the paper. Attached, you can find our point-by-point responses to your comments on the paper. To make the review process easier for reviewers, the revised parts related to your comments are marked in green.

The following is list of your comments with corresponding explanations:

  • Comment-No1: In section 2.3.1, it would be good to discuss briefly the statebased models like Markov, Semi-Markov and Petri Nets as well. In section 2.3.2, you have mentioned “Markov models in [35] and Petri nets are [36,37] are used to reduce the testing effort and dependability analysis.” However, I think they should be considered in section 2.3.1 not 2.3.2. Because the way the Markov and PN is used in Safety evaluation is actually based on historical data. Models like Hidden Markov Model (HMM) can be considered for Machine Learning as they has training procedure.
    • Correction: Thank you for your comment. We removed the sentence about
      the Markov model and Petri-net from Section 2.2.3 (old section 2.3.2) from line 296 and added it to section 2.2.2 (old section 2.3.1) in line 277. Also, we briefly explained the Markov and Petri-net in section 2.2.2 (old section 2.3.1) in line 278.
  •  Comment-No2: In section 2.3.2, also, the safety of ML algorithms themselves should be briefly discussed.
    • Correction: The safety of ML is an interesting topic, however, ML-based subsystems are often considered a black box. We consider fault injection in the interface between ML-based subsystems and other subsystems for safety assessment. To address your comment, we explained it at the end of the Introduction section in line 140.
  • Comment-No3: Figure 2 was very helpful to understand the idea behind the proposed method. However, it was a bit vague how the hyperparameter tuning is going to be performed. Could you please elaborate on that?
    • Correction: Thank you for your comment. We explained hyperparameter
      tuning more in line 384.
  • Comment-No4: In section 4.4, it is recommended to compare the results with other types of RL as well. For example, actor-critic and Deep RL could also have better results.
    • Correction: It is indeed necessary to show the difference between different
      types of RL algorithms. We already show in table 8 a comparison of different algorithms. However, the reason why this might have been overlooked is that the rationale for the selection of algorithms was not adequately pointed out in the experimental design of the paper. We, therefore, added a sentence to Section 4.3 in line 750 to explain which types of RL algorithms are compared. 

Kind regards,

Mehrdad

Reviewer 2 Report

Generic Review Comments:

The paper proposes an approach to the identification of  catastrophic faults based on model-level fault injection (FI) testing. The key idea is to facilitate FI testing by combining domain specific knowledge that captures the  system design and safety with reinforcement learning (RL) agent that infers and automatically selects the catastrophic faults to be injected during the FI testing. It is assumed such catastrophic faults occur likely far away from the central part of overall fault distribution (i.e. in the tail of long-tailed distribution) and therefore are costly to be treated properly. The claimed contributions include 1) a method for identifying dynamic (catastrophic) faults based on RL; 2) a method for assessing the safety risks due to failures in specific environmental conditions according to the related knowledge about the system configuration, behaviors and safety constraints; and 3) a specific guideline for configuring RL agents, including the reward functions, for FI testing.

The paper is largely well written with many merits regarding the review of related concepts and coverage of state-of-the-art approaches. However, the content fails to be sufficiently rich regarding the claimed contributions in the areas of fault modeling, dynamic fault injection and safety assessment. To be accepted as an article of Sensors, there would also be a need of emphasizing the fault behaviors and FI testing for sensory components or perception subsystem in automated-driving systems.

Specific Review Comments:

Line 40-41: “…However, most faults cause masked errors. These masked errors do not propagate further to the external state of the system.”

Line 43-44: “Traditional fault injection-based safety assessment methods (i) are not efficient in finding catastrophic faults (in the tail of the distribution) and do not scale up to the complexity of future systems.

Comment: RL could be effective in automating the exploration of fault/error space. The challenge of finding catastrophic faults has been shifted to the generation of trading data. A more explicit statement about how  such a data generation challenge is addressed would expected here.  

Line 45- 46: Future systems are consist of hardware such as  sensors, mechanical and hydraulic components, etc. as well as software for controlling or monitoring the hardware via transferring data and a high level of interaction.

Line 53- 54: Additionally, many of these systems are composed of complicated components (like heterogeneous systems), and the systems implementation details are not adequately  understood

Comment: It would be preferrable to elaborate how system hierarchy, composition and component complexity would affect the strategy of  fault injection and highlight the contribution of proposed approach.

Line 66- 71:  “a machine learning method to improve the efficiency of the model-implemented fault injection. Model-implemented fault injection enables us to front-load the safety assessment to the model level ….the proposed method examines the model under test and its environment for fault injection, instead of focusing on either.”

Comment: Properly trained RL model needs to have their “reward functions” defined according to the criticality of  consequent system failures. Is the mechanism for dynamic safety assessment clearly explained? How does the proposed approach establish and quantify the causal correlations among faults and failures (which are essentially given by the collected time-series data)  for the assessment of requirement fulfillment? How do the compositional effects of multiple faults treated?

Line 400-444: In this paper, e propose three alternatives to shape the reward function. The first alternative is to use the simulation models already defined evaluation functions…The second alternative is to transform the model’s robustness and safety requirements  into a logical formula to monitor the observation signals in the model….The third element alternative is to specify a custom reward shaping based on domain knowledge; therefore, the safety engineer identifies intended signals in the model under test and corresponding weights for each signal (to show the importance) to the RL agent. In our method, these signals must be represented as follows:…”

Comment: The concept “simulation models already defined evaluation functions” is not explained sufficiently. Could the robustness and safety requirements in logical formulas be related to the  “simulation models already defined evaluation functions” as well as the “custom reward shaping”.

Equation 3

Comment: Please elaborate necessary methods for quantifying of “strength”(sensitivity). How would the discretization be supported by the quantifications?  

Line 77- 80: Proper setup for fault injection experiments using the RL agent is challenging for safety engineers. As such, the safety engineer needs a proper guideline and workflow to configure the RL agent. We use domain knowledge about the model under test to reach the proper reinforcement learning configuration.

Line 82-84: “The reinforcement learning agent injects faults at each simulation step and eventually learns how to fail the model under test. The agent identifies fault patterns or a sequence of faults that causes a catastrophic failure in the  model under test. In essence, the agent learns to fail the system.

Line 133-140: “An essential element of fault injection simulation is the fault parameters. Based on [7], faults have three main parameters that create the fault space . ..”

Line 334-335: “Model architecture: Defines how the model is built up in terms of high-level functional blocks, how they are connected and transfer data, how the controllers work, how many states it has, etc. This helps determine those parts where faults should be injected.

Comment: An  RL based FI model can be applied to a component, a module or a subsystem, across the abstraction levels of system. How does the proposed approach support the selection of FI target(s)?

Dynamic fault conditions have spatial and temporal concerns, with for example multiple locations of injection. A learned RL model basically reflects the behaviors implied by the training data. The topology of system often contains loops (e.g. feeding outputs back to the inputs). How could such loops affect the learned RL model?

Line 109-110:  The proposed method injects fault in the signals within the model under test, and it can be applied to black-box models as long as signals are observable or reachable externally.

Comment: For a system component/module,  there could be primary and secondary faults according to their origins. Does the proposed method not consider internal faults (i.e. primary faults) in FI testing?

Line 364 -374: “The action signal is connected to the fault model and controls the fault parameter. The  RL agent makes different actions through fault space exploration and eventually converges to the desired sequence of actions that indicate a catastrophic fault. For each action the agent can make, we specify the following properties: Fault Range: A fault range must be specified, allowing the agent to define the boundaries of space exploration. These ranges help us scale the signal to [-1,1], mainly for performance reasons.

Line 444 – 459: “ 3.2.3. Fault Injection Models The fault models must be parameterized so that the RL agent can decide upon the value, time, and possibly place of the injected fault. These faults are often categorized as data changing faults and are added as disturbances to the connections between different parts of the model, e.g., sensors and controllers.

Comment:  It seems the selection of FI targets is dynamically done (“A subsystem or system of interest in the model for the fault injection simulation…”). What are the selection criteria and how the training and operation supporting this? 

These statements could be in contradiction with the experiment, which supports  only single location and single fault type. The experiment also shows only restricted failure mode (ways of data changing). See e.g. Line 594 -560 “In this paper, for simplicity,  we only simulate fault in a single location in the model under test in the velocity signal (first row of the table).”. More elaborations are preferable here.

Line  87-88: the reinforcement learning-based method for fault injection that utilizes the domain knowledge to set up the different parts of the reinforcement learning agent to find catastrophic faults in the model under test.

 Line 94-96: “Secondly, we focus on extracting high-level domain knowledge. This domain knowledge is based on the model under test (e.g., models architecture, inputs boundaries, and models states), safety requirements, and temporal behavior of the model under test.

Comment:  The configuration of RL agents for FI, regarding e.g. the locations, could be systematic or selective. How does the domain knowledge support the configuration?  

 Line 186-209: “2.1.3. System Requirements and Signal Temporal Logic ….”

Comment:  Safety requirements are often related to the probabilities of system failures and their criticalities, where soft errors of components are often of particular concern. Regarding the specification, a further elaboration about the probabilistic aspects of safety requirements and their impacts on the RL design is expected here.

Line 217-224 “…For example, they can automatically generate test suits for testing the software..”

Comment: It is preferrable to introduce how models support the generation of automated testing by mentioning test case generation (e.g. relating to the generation of scenarios, stimuli, …, etc.)

Line 235-249 2.3.2. Machine Learning …”

Comment: Further elaborations about key generic design steps of ML are needed. For example, feature engineering constitutes an important step in  ML. Could Principal component analysis (PCA) be applied for reducing the dimensions of high-dimensional data? How could time series data that constitute the system emissions by FI testing be treated?

Line 285 – 290 3. Reinforcement-based Catastrophic Fault Identification Method This section introduces a guideline to assist the user in setting up the RL agent for fault identification. The high-level idea of this method is to employ a reinforcement learning agent for model-implemented fault injection to find catastrophic faults in the model under test. Catastrophic faults violate the model’s specifications or safety requirements, and the proposed method semi-automatically identifies them.

“ Algorithm 1: …Line 6: Env FMU of the RL environment with Reward

Line 291-296:  The underlying neural net in the RL agent can understand the system functionality and control the experimentation. The RL agent injects fault in each step of the simulation, and interacts with both the controller and the environment around the controller. Therefore, fault identification is done in a broader context that matches the real-world case studies.

 Comment:  One key step of the proposed approach is related to the safety analysis regarding the risks of system failures. It is necessary to provide elaboration on how the safety analysis is automated while having the results encoded as reward functions in all three alternatives to shape the reward function (Line 400).

It is also worth to clarify if the interactions with environment affect the selection of operational scenarios.

Line 329-330: “Fault model: Specifying the model of fault that each has two parameters, such as the amplitude of the fault and activation time. “

Comment: What is  the exact definition of fault and Fault Model, instead of using “two parameters, such as…”? Would multiple faults be possible at specific time instants? How are the faults related to the signal boundaries and input/output ranges?

Other comments:

    Line 45: “Future systems are consist of hardware ….”

Comment: Typo.

    Line 216: “2.3. Test Automation Methods”

Comment: Should this be subsection 2.2.1 ?

    Line 319-320: “During the training, we log training data …. Afterward, we move to the training step,…” ´

Comment: Are these two sentences in wrong order?

Author Response

Dear Reviewer,

We take this opportunity to express our gratitude for your constructive feedback and helpful comments on improving our manuscript.
We have carefully studied the comments and corrected them in the paper. Attached, you can find our point-by-point responses to your comments on the paper. To make the review process easier for reviewers, the revised parts related to your comments are marked in blue.

The following is list of your comments with corresponding explanations:

  • Comment-No0: Genreric comment The content fails to be sufficiently
    rich regarding the claimed contributions in the areas of fault modeling, dynamic fault injection and safety assessment. To be accepted as an article of Sensors, there would also be a need of emphasizing the fault behaviors and FI testing for sensory components or perception subsystems in automated-driving systems.
    • Correction: It is indeed true that our contribution is an enabling technique usable in the context of fault behaviors and FI testing for sensory components or perception subsystems in automated-driving systems. We indeed failed to properly scope the contribution to the interest of the journal audience. To mend this, and appeal more to the readers of Sensors, we addressed this comment in the introduction section line 37-49.
  • Comment-No1: RL could be effective in automating the exploration of fault/error space. The challenge of finding catastrophic faults has been shifted to the generation of trading data. A more explicit statement about how such a data generation challenge is addressed would expected here.
    • Correction: This is indeed a good insight. To emphasize this, we added this in two separate parts in the introduction in lines 58 and 94.
  • Comment-No2: It would be preferable to elaborate on how system hierarchy, composition and component complexity would affect the strategy of fault injection and highlight the contribution of proposed approach.
    • Correction: This is a very interesting point raised by the reviewer. We addressed it in three parts of the paper. To address the effect of the system components on fault injection strategy, we added some sentences in line 71 and line 126 in the introduction. For addressing the effect of the system architecture on fault injection strategy, we added a paragraph in the discussion in line 945 as follows: There is indeed a typical hierarchy in the system because of processes/workflows for building these technical systems (e.g., V-model or other MBSE workflows). When designing the system, in early stages, you can use the models to already find different catastrophic faults. This information flow can be used as domain knowledge throughout the workflow (and further detailing) of the design, to better steer the fault-injection experiments. When moving to a more silo’ed design phase of the sub-systems and components, the system under test changes to the subsystem or component. Again information about this specific domain can be used as domain knowledge in the subsequent experiments. When integrating, the opposite should be done. Information about faults can propagate upwards to the system level and be used to identify catastrophic faults at that level. This raises some interesting challenges of automatically propagating this information between the different levels of abstraction. And also for designers, what can I tolerate as faults at each component/subsystem such that faults are masked. While this is a very interesting comment, we consider this out of the scope of this study. 
  • Comment-No3: Properly trained RL model needs to have their “reward functions” defined according to the criticality of consequent system failures. Is the mechanism for dynamic safety assessment clearly explained? How does the  proposed approach establish and quantify the causal correlations among faults and failures (which are essentially given by the collected time-series data) for the assessment of requirement fulfillment? How do the compositional effects of multiple faults treated?
    • Correction: Indeed the dynamic safety assessment was not clearly mentioned in the paper. To address this comment, we added some sentences in line 96. Also, multiple-point fault injection is out of the scope of our method, although it can be done.
  • Comment-No4: The concept of “simulation model’s already defined evaluation functions” is not explained sufficiently. Could the robustness and safety requirements in logical formulas be related to the “simulation model’s already defined evaluation functions” as well as the “custom reward shaping”.
    • Correction: In fact, we needed more explanation. We elaborated ”evaluation functions” more in line 489. Using the robustness and safety mechanism was already explained in the paper.
  • Comment-No5: Please elaborate necessary methods for quantifying of “strength” (sensitivity). How would the discretization be supported by the quantification?
    • Correction: Actually, this was not explained in the paper and thank you for highlighting it. We elaborated on it in line 533.
  • Comment-No6: An RL based FI model can be applied to a component, a module or a subsystem, across the abstraction levels of system. How does the proposed approach support the selection of FI target(s)? Dynamic fault conditions have spatial and temporal concerns, with for example multiple locations of injection. A learned RL model basically reflects the behaviors implied by the training data. The topology of system often contains loops (e.g. feeding outputs back to the inputs). How could such loops affect the learned RL model?
    • Correction: We clarified these comments in two parts. The first question is answered in line 390. For the second question, the internal structure of the model under test does not affect the RL training and it is elaborated in line 418.
  • Comment-No7: For a system component/module, there could be primary and secondary faults according to their origins. Does the proposed method not consider internal faults (i.e. primary faults) in FI testing?
    • Correction: Thank you for your comment. We further explained it in line 144. 
  •  Comment-No8: It seems the selection of FI targets is dynamically done (“A subsystem or system of interest in the model for the fault injection simulation. . . ”). What are the selection criteria and how the training and operation supporting this? These statements could be in contradiction with the experiment, which supports only single location and single fault type. The experiment also shows only restricted failure mode (ways of data changing). See e.g. Line 594 -560 “In this paper, for simplicity, we only simulate fault in a single location in the model under test in the velocity signal (first row of the table).”. More elaborations are preferable here.
    • Correction: For further clarification we added some sentences in line 539.
  • Comment-No9: The configuration of RL agents for FI, regarding e.g. the locations, could be systematic or selective. How does the domain knowledge support the configuration?
    • Correction: We explained it more in line 123.
  • Comment-No10: Safety requirements are often related to the probabilities of system failures and their criticalities, where soft errors of components are often of particular concern. Regarding the specification, a further elaboration about the probabilistic aspects of safety requirements and their impacts on the RL design is expected here.
    • Correction: Indeed probabilistic requirement are used often. We addressed it in line 938.
  • Comment-No11: It is preferable to introduce how models support the generation of automated testing by mentioning test case generation (e.g. relating to the generation of scenarios, stimuli, . . . , etc.)
    • Correction: This information was missing. We elaborated on it in line 264.
  • Comment-No12: Further elaborations about key generic design steps of ML are needed. For example, feature engineering constitutes an important step in ML. Could Principal component analysis (PCA) be applied for reducing the dimensions of high-dimensional data? How could time series data that constitute the system emissions by FI testing be treated?
    • Correction: Indeed it was not in that section. We addressed the first question in line 288 and second question in 298. 
  • Comment-No13: One key step of the proposed approach is related to the safety analysis regarding the risks of system failures. It is necessary to provide elaboration on how the safety analysis is automated while having the results encoded as reward functions in all three alternatives to shape the reward function (Line 400). It is also worth to clarify if the interactions with environment affect the selection of operational scenarios.
    • Correction: It is a good point that was missing in the text. For the first part of this comment, the safety analysis is automated in the workflow of the RL training and we explained it in line 483. The second part of the comment is elaborated in line 358.
  • Comment-No14: What is the exact definition of fault and Fault Model, instead of using “two parameters, such as. . . ”? Would multiple faults be possible at specific time instants? How are the faults related to the signal boundaries and input/output ranges?
    • Correction: We added some sentences with example to address your comment. The first question is addressed in line 403 and second question is addressed in line 412.
  • Comment-No15: Line 45: “Future systems are consist of hardware . . . .” Comment: Typo.
    • Correction: It has been fixed.
  • Comment-No16: Line 216: “2.3. Test Automation Methods” Comment: Should this be subsection 2.2.1 ?
    • Correction: Indeed, it was wrong. We fixed it.
  • Comment-No17: Line 319-320: “During the training we log training data . . . . Afterward, we move to the training step,. . . ” ´ Comment: Are these two sentences in wrong order?
    • Correction: Yes, there were in wrong order. We fixed it. 

Kind regards,

Mehrdad

Reviewer 3 Report

Abstract

- good. well written.

Introduction.

- Line 43, I think removing the word Problem description is better. From the explanation, we know that the authors are discussing the problem related to the study.

- Line 65, remove 'Proposed method'.

- Line 86, remove 'Contributions'.

Background and Related Works

- In section 2.2, there is no related works discussion in this section.

Reinforcement-based Catastrophic Fault Identification Method

- this section was well written. good explanation.

Experimental Study

- Case studies, ACC is too old. suggestion use the latest case study. Add a link where the reader can refer to the case study. I'm not sure about AEB. is this a real case study? not mentioned in the manuscript. just state that has taken from MathWroks.

Discussion

- ok

Conclusion

- ok

Author Response

Dear Reviewer,

We take this opportunity to express our gratitude for your constructive feedback and helpful comments on improving our manuscript.
We have carefully studied the comments and corrected them in the paper. Attached, you can find our point-by-point responses to your comments on the paper. To make the review process easier for reviewers, the revised parts related to your comments are marked in red.

The following is list of your comments with corresponding explanations:

  •  Comment-No1: Line 43, I think removing the word Problem description is
    better. From the explanation, we know that the authors are discussing the
    problem related to the study.
    • Correction: Thank you for your comment. We removed it.
  • Comment-No2: Line 65, remove ’Proposed method’.
    • Correction: Thank you for your comment. We removed it.
  • Comment-No3: Line 86, remove ’Contributions’.
    • Correction: We removed it.
  • Comment-No4: In section 2.2, there is no related works discussion in this section.
    • Correction: Indeed it was missing in the text. We added some sentences in line 332.
  • Comment-No5: Case studies, ACC is too old. suggestion use the latest case study. Add a link where the reader can refer to the case study. I’m not sure about AEB. is this a real case study? not mentioned in the manuscript. just state that has taken from MathWroks.
    • Correction: Actually, the ACC case study is very intuitive. It is widely being used in articles and easy to understand for readers. That is why we explained our algorithm based on it. The second case study is elaborated in the supplementary file in Zenodo, therefore the future readers can refer to it. We added some sentences in three part such as line 631, 657 and 668. In those sentences, we clarified the ACC and AEB are examples from MathWorks representing the real-world functionality of them. 

Kind regards,

Mehrdad

Round 2

Reviewer 2 Report

After the review iteration, the authors addressed the issues being pointed out. I consider the current version has reached an acceptable quality level.